# Genetic Diversity and Selective Signature in Dabieshan Cattle Revealed by Whole-Genome Resequencing

**DOI:** 10.3390/biology11091327

**Published:** 2022-09-08

**Authors:** Xiwen Guan, Shuanping Zhao, Weixuan Xiang, Hai Jin, Ningbo Chen, Chuzhao Lei, Yutang Jia, Lei Xu

**Affiliations:** 1Anhui Province Key Laboratory of Livestock and Poultry Product Safety Engineering, Institute of Animal Husbandry and Veterinary Medicine, Anhui Academy of Agricultural Sciences, Hefei 230031, China; 2Key Laboratory of Animal Genetics, Breeding and Reproduction of Shaanxi Province, College of Animal Science and Technology, Northwest A&F University, Xianyang 712100, China; 3School of Biological Science, University of Bristol, 24 Tyndall Avenue, Bristol BS8 1TQ, UK

**Keywords:** whole-genome sequencing, population structure, diversity, genetic signatures, indigenous cattle breeds

## Abstract

**Simple Summary:**

To protect the genetic resources of Chinese native cattle breeds, we investigated the genetic structure, genetic diversity and genetic signature from artificial or natural selection by sequencing 32 bovine genomes from the breeding farm of the Dabieshan population. We discovered that the ancestral contributions of Dabieshan originated from Chinese indicine and East Asian taurine on the autosomal genome, which had abundant genomic diversity. Some candidate genes associated with fertility, feed efficiency, fat deposition, immune response, heat resistance and the coat color were identified by a selective sweep. The SNPs data were based on genomics, which could establish a foundation for breed amelioration and support conservation for indigenous cattle breeds.

**Abstract:**

Dabieshan cattle are a typical breed of southern Chinese cattle that have the characteristics of muscularity, excellent meat quality and tolerance to temperature and humidity. Based on 148 whole-genome data, our analysis disclosed the ancestry components of Dabieshan cattle with Chinese indicine (0.857) and East Asian taurine (0.139). The Dabieshan genome demonstrated a higher genomic diversity compared with the other eight populations, supported by the observed nucleotide diversity, linkage disequilibrium decay and runs of homozygosity. The candidate genes were detected by a selective sweep, which might relate to the fertility (*GPX5*, *GPX6*), feed efficiency (*SLC2A5*), immune response (*IGLL1*, *BOLA-DQA2*, *BOLA-DQB*), heat resistance (*DnaJC1*, *DnaJC13*, *HSPA4*), fat deposition (*MLLT10*) and the coat color (*ASIP*). We also identified the “East Asian taurine-like” segments in Dabieshan cattle, which might contribute to meat quality traits. The results revealed by the unique and valuable genomic data can build a foundation for the genetic improvement and conservation of genetic resources for indigenous cattle breeds.

## 1. Introduction

Cattle (*Bos taurus*) were amongst the first livestock species domesticated by humans [1]. Approximately 10,000 years ago, wild aurochs (*Bos primigenius*) began to be domesticated by humans [2], which were dispersed worldwide with the development of crops. Currently, more than 1000 breeds and varieties have been identified worldwide [3]. Based on habits and morphological characteristics, *Bos taurus* can be classified as two subspecies, *B. t. taurus* (humpless taurine) of European origin, and *B. t. indicus* (humped indicine) of Indian origin [4]. There are 55 domestic cattle breeds in China, which are valuable genetic resources to promote cattle breeding under changing production and environmental conditions in the future [5].

Dabieshan cattle (population head size approximately 300,000–400,000) are one of the typical Chinese native cattle breeds, residing in the surrounding areas of the Dabie Mountains and the middle and lower reaches of the Yangtze River, and are mostly used for draft and beef production. The breeding scale of Dabieshan cattle in scatter-feed is approximately 10–20. The breeding method is mainly barn feeding, supplemented by grazing, and the system of reproduction is mating in a group as a minimum, so Dabieshan in scatter-feed have a high degree of inbreeding. The cattle farming records of the region can be tracked back to as early as approximately 500 A.D., and were identified by the Anhui Agriculture Ministry in 1982 [5]. The bovine production value is an important standard for breeding, resulting in the characteristics of muscularity, roborant skeletons and the small body size of Dabieshan cattle. With the development of modern agriculture, this breed has been considered as a source of beef due to its excellent meat quality and tough feeding resistance [5]. Previous studies using mitochondrial DNA analysis showed that Dabieshan cattle had two types of maternal origin, from Bos taurus and Bos indicus, and the displacement loop regions had rich genetic diversity (π = 0.023) [6]. The Y-SNPs and Y-STRs markers indicated that this breed belongs to the Southern cattle type, with such limited paternal genetic diversity in China, showing the single and stable genetic basis of the germplasm characteristics of indicine [7]. At present, research on Dabieshan cattle is limited to the mitochondrial and Y chromosome, whereas available WGS data provides results such as population structure, genetic diversity and selection pressure.

The investigations on cattle based on WGS data primitively focused on economically relevant traits for commerce. Subsequently, researchers have gradually focused their research on discovering adaptable indigenous breeds, such as thermal stress and immune responses in African cattle [8] or cold resistance in Yanbian cattle [9]. This change in focus can identify the genomic variation characteristics of environmental adaptability regarding indigenous cattle to some extent, which could offer a reference for designing genetic breeding strategies. Nevertheless, there has been no previous study to understand the more in-depth selection pressure, population structure and genetic diversity using WGS data in the Dabieshan breed. Therefore, in the present study, we obtained SNPs based on the *Bos taurus* reference genome from Dabieshan cattle in order to magnify our information about the genomic characteristics and selection pressure signals of natural and artificial selection in Dabieshan cattle of China

## 2. Materials and Methods

### 2.1. Ethics Statement

In our study, the experimental procedures were approved by the Experimental Animal Management Committee (EAMC) of Northwest A&F University (2011-31,101,684) and complied with the National Standard of Laboratory Animals Guidelines for Ethical Review of Animal Welfare (GB/T 35892-2018) and Guide for the Care and Use of Laboratory Animals: Eighth edition.

### 2.2. Sample Collection and Genome Sequencing

This study randomly selected ear tissue samples of Dabieshan cattle (*n* = 30) that had an age of approximately 3 years from the core breeding farm in Taihu County of Anhui Province (Appendix A). The genomic DNA was extracted using the phenol-chloroform method [10]. We constructed two libraries with insert sizes of 500 bp using a 2 × 150 bp model, and the whole genome resequencing was performed using Illumina NovaSeq instruments (Beijing, China). Furthermore, we collected the whole genome resequencing data of Dabieshan cattle (*n* = 2) and 116 representative breeds, including the five “core” cattle populations: Chinese indicine (Guangfeng cattle (*n* = 4), Ji’an cattle (*n* = 4), Leiqiong cattle (*n* = 3), Wannan cattle (*n* = 5)), Indian indicine (Brahman (*n* = 4), Gir (*n* = 2), Hariana (*n* = 1), Nelore (*n* = 1), Sahiwal (*n* = 1), Tharparkar (*n* = 1), unknow (*n* = 1)), East Asian taurine (Hanwoo (*n* = 15)), European taurine (Hereford (*n* = 10), Red Angus (*n* = 5), Angus (*n* = 10)) Eurasian indicine (Simmental (*n* = 8), Jersey (*n* = 11)) and Qinchuan cattle (*n* = 30) (Appendix A).

### 2.3. SNP Calling

At first, the cleaned reads were mapped to the *Bos taurus* reference genome assembly (ARS-UCD1.2) using BWA-MEM (v0.7.13-r1126, RRID:SCR_010910) [11], with an average mapping rate of 99.71%. We filtered potential duplicate reads by Picard tools (http://broadinstitute.github.io/picard, accessed on 15 November 2020) (REMOVE_DUPLICATES = true) [12]. Then, we used the Genome Analysis Toolkit (GATK 3.8, RRID:SCR_001876) [12] to detect SNPs. The criteria implemented for all SNPs were as follows: (1) QD (quality by depth) < 2; (2) MQ (mapping quality) > 40.0; (3) FS (Phred-scaled *p* value using Fisher’s exact test to detect strand bias) < 60.0; (4) MQRankSum (Z score from the Wilcoxon rank sum test of Alt vs. Ref read MQs) > −12.5′; (5) ReadPosRankSum (Z score from the Wilcoxon rank sum test of Alt vs. Ref read position bias) > −8.0; and (6) DP (mean sequencing depth) < 1/3× and > 3×. Finally, we used the table_annovar.pl module of ANNOVAR [13] (RRID:SCR_012821) to annotate the distribution of SNPs within various genomic regions and used SnpEff [14] (RRID:SCR_005191) to calculated Ts/Tv.

### 2.4. Population Genetic Structure

The SNPs with high levels of pairwise LD were pruned by PLINK [15] with the parameter (--indep-pair-wise 50 5 0.2). We constructed a phylogenetic tree with the neighbor-joining (NJ) method using PLINK software [15] (RRID:SCR 001757) with the matrix of pairwise genetic distances, which aims to establish the evolutionary relationship among 148 cattle. The NJ tree was visualized by MEGA v5.0 (RRID:SCR_000667) [16] and FigTree v1.4.4 (http://tree.bio.ed.ac.uk/software/figtree/, accessed on 5 July 2021). The principal component analysis (PCA) of SNPs was conducted using the smart PCA program of EIGENSOFT v5.0 [17] (RRID:SCR_004965). This study used ADMIXTURE v1.3.0 (RRID:SCR_001263) [18] for K = 4 to explore the global ancestral proportions of Dabieshan cattle.

### 2.5. Genomic Diversity

In order to reduce the experimental error caused by the number of animals, we randomly selected 15 Dabieshan cattle and 15 Qinchuan cattle. The nucleotide diversity of each breed was carried out in 50 kb windows with 50 kb steps using VCFtools [19] (RRID:SCR 001235). Using PopLDdecay software [20] (RRID:SCR_022509) with default parameters, we estimated linkage disequilibrium (LD) decay with the physical distance between pairwise SNPs. The length and number of runs of homozygosity (ROH) per individual were calculated using VCFtools [19] with the following parameters: —homozyg-density 50—homozyg-window-het 3—homozyg-window-missing 5. Based on four categories (0.5–1 Mb, 1–2 Mb, 2–4 Mb, >4 Mb), the ROH of each breed was classified.

### 2.6. Genome-Wide Selection

The selective sweep in genomics of Dabieshan cattle was calculated using the following strategy. First, the positive selection regions within Dabieshan cattle were computed by the nucleotide diversity (θπ) and the composite likelihood ratio (CLR) [21]. The θπ analyzed the diversity of the complete polymorphism data using VCFtools with 50 kb window size and 20 kb step size [19]. The CLR checks were estimated using SweepFinde2 [22] for sites within a non-overlapping 50 kb window. The top 1% windows of each method were considered as candidate regions. The Tajima’s D statistic was performed by using VCFtools for the candidate gene.

Secondly, we performed comparisons between Dabieshan cattle versus Hanwoo using the fixation index (*F*_ST_) [23], genetic diversity ratio (θπ-Ratio) [23] and cross-population extended haplotype homozygosity (XP-EHH) [24] for revealing genetic variants related to the adaptation of Dabieshan cattle. The *F*_ST_ value, a method of statistical selective sweep regions based on population differentiation, is generally used to measure the reduction in heterozygosity of the target population relative to the reference population. θπ-ratio is the ratio of θπ values between the different populations of one species, and represents the degree of differentiation. We used VCFtools to perform *F*_ST_ and θπ-ratio analysis [19] with 50 kb sliding and 20 kb step. For the XP-EHH, selscan v1.3.0 [25] was used to calculate the positive selection signals of each population in each 50 kb window. The top 1% (*F*_ST_ > 0.58, θπ-ratio > 0.24, XP-EHH > 2.45) was chosen as the significance threshold, and genes in those regions were defined as potential candidate genes. We performed comparisons between Dabieshan cattle versus Qinchuan cattle using the *F*_ST_, which is consistent with the previous approach. The top 1% windows (*F*_ST_ > 0.38) were deemed to be in the candidate region.

Finally, we applied RFMix v2.03-r0 [26] to perform East Asian taurine ancestry inference on autosomes of the Dabieshan cattle genome. We selected the genome of Hanwoo as the reference group in order to calculate “East Asian taurine-like” segments.

### 2.7. Functional Prediction Analysis

Gene ontology (GO) biological processes (BP) and Kyoto Encyclopedia of Genes and Genomes (KEGG) pathways were enriched using KOBAS 3.0 (http://bioinfo.org/kobas/, accessed on 5 March 2022) [27], which were significant when the corrected *p*-value was less than 0.05 for the understanding of the biological functions and signal transduction pathways of candidate genes in Dabieshan cattle.

## 3. Results

### 3.1. Whole-Genome Sequencing, Assembly, and Genetic Variation

Thirty samples generated an ~9.10 × coverage each (Appendix A), and then two additional WGS data of Dabieshan cattle were generated from Chen et al. [28]. A total of 42,407,831 biallelic SNPs were identified in 32 Dabieshan cattle (Appendix A). Figure 1a illustrates the functional classification of the polymorphic sites of 32 Dabieshan cattle. Most SNPs existed in intergenic (59.00%) and intron (38.05%) regions, and the remaining SNPs were present in the downstream (0.63%) and upstream (0.58%) regions of the open reading frame, untranslated regions (0.98%) and exonic regions (0.76%) that contained 120,972 non-synonymous SNPs and 192,147 synonymous SNPs.

In ordered to estimate the diversity of Dabieshan cattle, a tripartite combined genotyping of 32 Dabieshan cattle, the public genomes from 5 “core” cattle populations and Qinchuan cattle (Appendix A) was performed using GATK. In total, ~44 billion reads were generated, and the average alignment rate and sequencing depth reached 99.38% and ~11.70×, respectively. The transition-to-transversion (Ts/Tv) ratios were 2.344–2.408 [29,30], indicating the good quality of our SNP data (Appendix A). Furthermore, we performed a statistical analysis of SNPs and breed-specific SNPs based on the reference genome (ARS-UCD1.2). While comparing to other breeds, the number of SNPs and specific SNPs from taurine was significantly lower, which is consistent with previous studies [28]. Interestingly, the largest number (*n* = 42,407,831) and specific (*n* = 5,923,609) SNPs were found in the Dabieshan breed (Appendix A), which might be partly explained by the different sample sizes per breed and partly by breed differences.

### 3.2. Population Structure

In order to explore the population structure of Dabieshan cattle, we assessed the neighbor-joining (NJ) tree, principal component analysis (PCA) and ADMIXTURE using the autosomal genome data. In PCA (Figure 1b), the first component separated *Bos taurus* from *Bos indicus*, explaining approximately 6.03% of the total genetic variation information, which indicated considerable genetic distance between taurine and zebu. In PC2, Chinese indicine, Indian indicine, East Asian taurine and European-Eurasian taurine were effectively separated, explaining approximately 2.37% of the total genetic variation information. The third component explained 1.80% of the total variation and could not separate any breed (Appendix A). Moreover, the NJ tree (Figure 1c) demonstrated that five “core” cattle populations formed their own separate clusters. Dabieshan cattle had the closest genetic distance from Chinese indicine, which is consistent with the PCA results. These results were also repeated by ADMIXTURE analysis [31] (Figure 1d). The results displayed that, at K = 2, 149, animals could be basically distinguished from indicine and taurine. At K = 4, Dabieshan cattle shared genome ancestry with Chinese indicine (0.857), East Asian taurine (0.139) and European taurine (0.004).

### 3.3. Patterns of Genomic Variation

The patterns of genomic variation in the Dabieshan population were based on the analysis of nucleotide diversity, linkage disequilibrium (LD) decay and runs of homozygosity (ROH) with a window scale of 50 kb. In Figure 2a, we observed that the average LD (r^2^) was the highest in Jersey cattle, followed by Hereford cattle, Simmental cattle and Angus cattle, when the distance between adjacent SNPs was short (<10 kb), and, on the other hand, that the lowest average LD was found for Dabieshan cattle, then Qinchuan cattle and then followed by Chinese indicine and Indian indicine. The nucleotide diversity of Dabieshan (mean θπ = 0.0034) was lower than that of Chinese indicine cattle (0.0037) but approximately three times higher than that of European breeds (0.0007–0.0010). The tendency of nucleotide diversity (Figure 2b) was nearly consistent with those from LD decay, in which, the highest genome-wide genetic diversity was found in indicine cattle (Indian and Chinese indicine) and Dabieshan cattle.

In this study, the length of ROH was segmented into four size classes: 0.5–1 Mb, 1–2 Mb, 2–4 Mb and >4 Mb, for evaluating the ROH pattern of Dabieshan and other cattle breeds. Figure 2c shows that the majority of ROHs were found between 0.5–1 Mb, whereas, sensibly, European commercial breeds had more medium-sized (2–4 Mb) and long ROHs (>4 Mb). In addition, Dabieshan cattle presented a lower length of ROH compared to European commercial breeds, which had a greater length (Figure 2d). Consanguineous mating leads to the generation of long ROH, whereas shorter ROH mirror distant ancestral influences [32]. As expected, neglecting the sample size, the number and length of ROH were generally higher in taurine than in hybrid and indicine, suggesting that commercial breeds might have a risk of a decline in inbreeding.

### 3.4. Candidate Genes under Selective Sweep

The nucleotide diversity analysis (θπ) and the composite likelihood ratio (CLR) were employed to distinguish the positive selection region in the Dabieshan breed alone. Outlier signals (top 1%) were considered as candidate selective regions (Figure 3a). Based on the θπ test statistic, we obtained 1103 putative genes (Appendix A), whereas 529 positively selected genes were detected by the CLR test (Appendix A). These genes included *GPX5*, *GPX6* (reproductive performance) [33], *SLC2A5* (feed efficiency) [34,35,36], *IGLL1* [37] and *BOLA-DQA2*, *BOLA-DQB* (immunity) [8,38,39,40]. It is worth noting that the *MLLT10* gene, which is associated with fat deposition [41], was detected in both the two methods, indicating that the mentioned genes were strongly selected in Dabieshan cattle. The positive selection signals around the region of *MLLT10* (BTA13: 22.74–22.95 Mb) were further recapitulated by significantly lower values of nucleotide diversity and Tajima’s D in Dabieshan cattle (Figure 3b). To better understand the biological implications of the selection signals, we performed functional enrichment analysis for genes with selection signatures (Appendix A). A gene set enrichment analysis of KEGG pathways revealed that the most significant pathway was the “thyroid hormone signaling pathway” (corrected *p* value = 0.004, Appendix A), which is related to heat resistance. Over-representation analysis of GO BP terms showed that Dabieshan cattle had increased GO categories involved in “embryonic skeletal system morphogenesis” (corrected *p* value = 2.467 × 10^13^) and “ATP binding” (corrected *p* value = 1.202 × 10^6^) (Appendix A).

We divided the genome into non-overlapping segments of 50 kb [42] to compare the genomes of Dabieshan cattle to distinguish between East Asian taurine (Hanwoo) signatures of positive selection following environmental and human selection pressures. Outlier regions (the top 1% *F*_ST_, θπ ratio or XP-EHH statistics) were deemed to be candidate regions for further analysis. Based on *F*_ST_, we obtained 884 candidate genes (Appendix A). In contrast, 757 and 600 candidate genes were identified using the θπ ratio and XP-EHH (Appendix A) separately. Some of the candidate genes were outstanding genes related to heat resistance (*DnaJC1*, *DnaJC13*, *HSPA4*) [43,44,45,46], so they had a strong positive selection with respect to the adaptive characteristics of Dabieshan cattle (Figure 3c). By calculating *F*_ST_ with a smaller window (5 kb) and haplotype patterns, significant differentiation was observed between Dabieshan cattle and Hanwoo (Figure 3d). Synchronously, a missense mutation (p.Pro522Leu) was observed in the heat-resistance-related gene *DnaJC1* (Figure 3e). Allele G displayed a rare distribution (frequency 0.172) in Dabieshan cattle, whereas it showed an opposite pattern (frequency 1.000) in Hanwoo. Therein, the *HSPA4* gene was overlapped by three analyses. We also found one missense mutation that encodes a p.Leu792Pro substitution, and that the homozygous mutation (CC) and heterozygous mutation (CT) were highly conserved in the Dabieshan population (0.97) (Appendix A). Moreover, we noticed a region (25.58–25.63 Mb) scanned by the θπ ratio on chromosome 23 (including *BOLA-DQA2* and *BOLA-DQB*), illustrating a strong positive selection signal (Figure 3c). In the functional prediction of all candidate genes, significant KEGG pathways and GO BP terms (corrected *p* < 0.05) were mainly enriched in immunity (“immune response, GO:0006955”), heat resistance (“adrenergic signaling in cardiomyocytes, bta04261”, “thyroid hormone signaling pathway, bta04919”) and coat color (“melanogenesis, bta04916”) (Appendix A and Appendix A). The coat color of 32 Dabieshan cattle was black or gray-black. We tried to investigate the black or gray-black color of Dabieshan cattle by comparing them with Qinchuan cattle (reddish-brown). A significant differentiation of the *ASIP* gene was identified (*F*_ST_ = 0.4695), which was further verified by haplotype patterns (Appendix A). Previous research has suggested that the variant at the *ASIP* locus causes coat color darkening in Nellore cattle [47], which could be linked to the coat color variation in Dabieshan cattle.

Finally, we calculated 6,967,850 “East Asian taurine-like” segments in Dabieshan cattle, and a total of 1207 genes were identified in high-frequency regions (>0.9) (Figure 4a, Appendix A). Notably, from the DAVID gene ontology, there were some significant GO BP pathways related to muscle and skeleton development, such as “actin cytoskeleton, GO:0015629”, “actin filament network formation, GO:0051639”, “calcium ion binding, GO:0005509” and “BMP receptor binding, GO:0070700” (Figure 4b, Appendix A).

## 4. Discussion

Since the 1980s, a large number of exotic commercial cattle breeds have been introduced to blindly improve Chinese local cattle, leading to the disintegration of the excellent gene pool due to crossbreeding, and the genetic resources of Chinese local cattle are gradually being depleted. In order to protect the precious genetic resources of Chinese native cattle, it is crucial to design scientific and practical breeding programs that can contribute to the conservation of local Chinese cattle. Genomic data could reflect a wide variety of historical developing events, including climate adaptation, species introductions and artificial selection [48]. Thus, the exploration of population genetic structure and genetic diversity can be regarded as essential for genetic evaluation, reflecting recent cross-breeding and the utilization or conservation of genetic resources. The evaluation of population genetic structure and genetic diversity using mitochondrial DNA and Y chromosome markers is less precise than the assessment from the WGS data; therefore, the data obtained by WGS were adopted in this experiment.

A previous WGS analysis of 49 worldwide cattle breeds suggested that the domesticated cattle were divided into five “core” populations: Chinese indicine, Indian indicine, East Asian taurine, European taurine and Eurasian indicine [28]. In this study, we employed the five “core” cattle populations and an outstanding Chinese local breed as a reference group to explore the population genetic structure and genetic diversity of Dabieshan cattle. Dabieshan cattle are hybrid cattle, the ancestral contributions of which originated from Chinese indicine (85.7%) and East Asian taurine (13.9%), as shown in the ADMIXTURE analysis (Figure 1d). The nucleotide diversity, LD decay and ROH results are consistent with prior studies [49], directly indicating that the genetic diversity was higher and that inbreeding depression was lower compared to European commercial varieties. Chinese indicine had a high nucleotide diversity on the autosomal genome, which may be a prior reception of introgression from Banteng (*Bos javanicus*) [28]. The relatively abundant genomic diversity was found in Dabieshan cattle, which may result from hybridization with taurine and indicine. At the same time, populations of Dabieshan and Chinese indicine inhabit small farms and undergo constant selection and elimination with less inbreeding. The monotonous feeding environment and high degree of inbreeding may be responsible for the lower nucleotide diversity of European commercial breeds than indicine and hybrid cattle [50]. However, deleterious variants might reach a high frequency in European commercial breeds due to the frequent utilization of individual carrier animals in artificial insemination [51] or hitchhiking with favorable alleles under artificial selection [52,53], and even due to missing superior genes from stress-resistance traits such as disease resistance and heat stress tolerance.

In comparison to SNP arrays, we were able to identify more selective signatures using WGS data [54,55]. Using complementary approaches, we discerned several candidate regions for superior traits that were potential targets of artificial and natural selection in Dabieshan cattle.

Considering that Dabieshan cattle express a better performance in adapting to a high-temperature and high-humidity environment [5], further comparisons were made between Dabieshan and commercial breeds (Hanwoo). The *DnaJC1* gene (*F*_ST_ = 0.86) translates an endoplasmic reticulum heat shock protein that binds the molecular chaperone HSPA5 (alias BiP) [43]. The *DnaCJ13* gene-encoded protein, playing a role in clathrin-mediated endocytosis, is associated with the heat-shock protein Hsc70 [44]. The heat-shock protein family A (Hsp70) member 4 (*HSPA4*) is notable for promoting cell protection against heat damage and preventing protein denaturation [45,46]. Our KEGG analysis revealed two significant heat-stress-related pathways (“adrenergic signaling in cardiomyocytes” and “thyroid hormone signaling pathway”) (Appendix A). The epinephrine was elevated with prolonged environmental heat stress [56]. Meanwhile, thyroid hormones adjust the metabolic rates to facilitate the body thermal balance, which is quite important for heat adaption [57]. These results might have rapidly evolved in Dabieshan cattle, which might give a reasonable explanation for their utterly different degree of thermotolerance compared to the commercial breed. Dabieshan cattle are generally more resistant to disease than their commercial counterparts [5]. Many of our candidate regions harbored genes implicated in the immune system, and major histocompatibility complex (MHC) genes were annotated (*BoLA*, *BOLA*, *BOLA-DQB*, *BOLA-DQA2* and *BOLA-DQA5*). The MHC region contains a diverse array of genes that are crucial for the initiation of adaptive immune responses that map to chromosome 23 in cattle [38]. Notably, extensive studies have reported the importance of the bovine lymphocyte antigen complex in host immunity [39] and its correlation with parasitic diseases [8,40]. The *GBP4* and *GBP6* gene were detected in θπ and the θπ-ratio. GBPs mediate resistance against invading pathogens [58]. In the functional enrichment of all candidate genes, the significant “immune response, GO:0006955” pathway (corrected *p* = 0.01) was enriched (Appendix A and Appendix A).

Bovine fecundity is a crucial factor for a successful reproductive performance; as for Dabieshan cattle, they possess early sexual maturation (10–13 months) [5]. In the glutathione peroxidases (GPxs) gene family, the GPX5 protein plays a vital role in the maintenance of spermatozoa DNA integrity [59]. The *GPX6* is regarded as an essential participator in the in vitro induced capacitation and acrosome reaction in porcine sperm [33]. Feed efficiency is an important economic factor affecting beef production costs [60]. It has been reported that Dabieshan cattle have the characteristic of roughage resistance and have a high potential to produce highly marbled beef [5]. Similarly, Dabieshan cattle as tillage livestock have the typical characteristics of muscularity and roborant skeletons, with intense artificial selection. A large amount of evidence shows that the *SLC2A5* gene is involved in the metabolism of carbohydrates [34], and the increased expression of *SLC2A5* has been associated with fat deposition [35]. The increased *SLC2A5* expression was associated with an enhanced relative growth rate and Kleiber index in Nelore steers [36]. The copy number variation of the *MLLT10* gene sensibly influenced the growth traits of two Chinese native cattle breeds [41]. Similarly, the *MLLT10* gene had a feed conversion ratio (FCR) regarding male Duroc pigs [61]. Furthermore, the functions of these KEGG and GO pathways from the “East Asian taurine-like” segments may contribute to fat deposition and muscle and skeletal development (Figure 4b). Based on GeneCards annotation (https://www.genecards.org/, accessed on 10 May 2022), the *ROCK2* gene regulates the formation of actin stress fibers, which is related to intramuscular fat under selection in Ankole cattle [62]. Through the activation of the mTOR signaling pathway, an increase in the level of *bone morphogenetic protein 7* (*BMP7*) transcription or activity of its receptor in muscle cells can induce hypertrophy [63]. On the contrary, the BMP4 growth factor increases the number of activated satellite cells and regulates muscle regeneration [64,65]. Moreover, the *PPP3R1* gene encodes a phosphatase protein, and the expression levels of the *PPP3R1* gene may result in differences in muscle deposition in Nelore cattle [66]. The *PPP3CA* gene has been shown to be involved in the differentiation of perimuscular preadipocytes in cattle [67], which are responsible for the storage of fat [68]. A famous beef meat grade, A4 (the Japanese Beef Meat Grade scale, 2008), can be produced by purebred Dabieshan cattle with nutritional regulation. Grade A4 is recommended as the most highly satisfactory and most economical top grade of fattened beef for Chinese consumers [69], which have tremendous market potential. Because of these genes and pathways, we hypothesize that Dabieshan cattle have the potential to become a source of high-quality beef production.

Thirty-two Dabieshan cattle from the core breeding farm have purified and uniform ancestral contributions with a small impact of globalization, which is suitable for breed conservation and the in-depth exploration of genetic resources. We highlighted some adaptations at the genome level, which may relate to climatic challenges (e.g., heat and disease) and artificial selection (e.g., fertility, fat deposition, fat deposition, coat color). Some of our results might provide deep insight toward enhancing livestock production and increasing adaptation to local environmental challenges through crossbreeding (e.g., marker-assisted selection to increase haplotype frequency). In practice, these results may develop new genomic evidence for the design and implementation of improved poultry productivity. The Dabieshan cattle are predominantly fed in small groups (approximately 10–20) by farmer. Small group feeding affects a high degree of inbreeding, which can lead to the inhibition of various crucial traits. At the same time, the reproduction rate and productivity are unstable due to extensive feeding. The breeding efficiency is low under the traditional breeding mode; the breeding stock gradually declines. At present, there are many native cattle breeds in the world that have excellent traits, and not only Dabieshan cattle, but they are in a state of scattered breeding, and the sufficient protection and utilization of their genetic resources need to be investigated.

## 5. Conclusions

Our study presented a comprehensive overview of the Dabieshan genome by applying whole genome sequencing. Dabieshan cattle originated from Chinese indicine and East Asian taurine and demonstrated abundant genomic diversity. Additionally, the highlighted regions and genes that provide evidence of past and ongoing selection in Dabieshan cattle are enriched for candidate genes that may be associated with immune response, heat stress, reproduction, feed efficiency and meat quality traits. These results extend the application for further studies on the genomic characteristics, genetic resource evaluation and conservation strategies for Chinese native cattle breeds.

## Figures and Tables

**Figure 1 biology-11-01327-f001:**
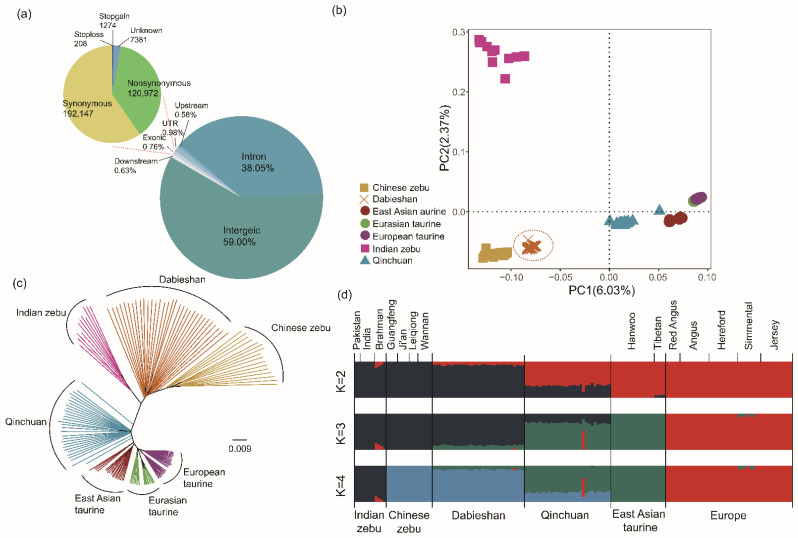
Population genetic structure and relationship among Dabieshan cattle using autosomal variants. (**a**) An NJ tree was constructed on the basis of 148 animals. (**b**) PCA showing PC 1 against PC 2. (**c**) Distribution of ANNOVAR annotation SNPs with whole-genome data. (**d**) Population genetic structure of cattle breeds using ADMIXTURE with K = 4.

**Figure 2 biology-11-01327-f002:**
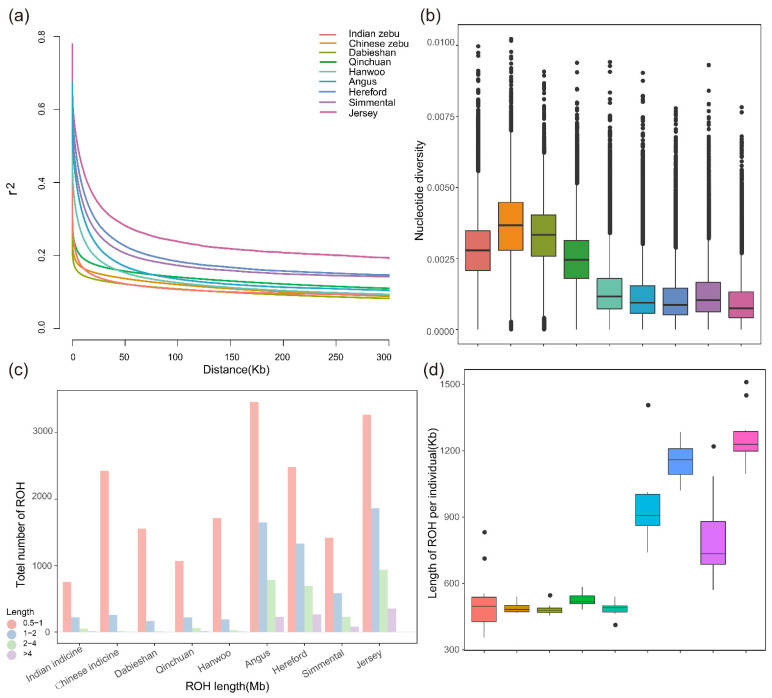
Genetic diversity among nine population. (**a**) The average LD decay calculated per group on whole-genome autosome. (**b**) Nucleotide diversity at a whole-genome scale of each group. The horizontal line inside the box indicated the median of this distribution and dots can be considered as outliers. (**c**) The distribution of lengths ROH in four categories: 0.5–1, 1–2, 2–4, >4 Mb. (**d**) The length of ROH estimated from each population.

**Figure 3 biology-11-01327-f003:**
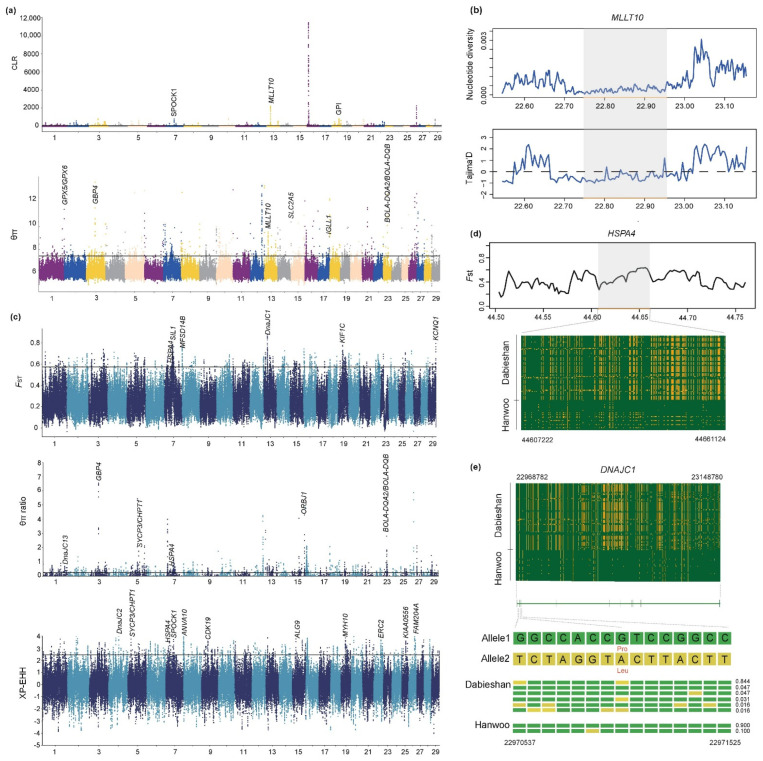
The signatures of selective sweep in Dabieshan cattle. (**a**) Manhattan plot of CLR and θπ for the autosomes. The line indicates the threshold line for the top 1% region (**b**) The line chart of nucleotide diversity and Tajima’s D at the *MLLT10* gene region. (**c**) Manhattan plot of *F*_ST_, θπ-ratio and XP-EHH based on Hanwoo. (**d**) *F*_ST_ and haplotype sharing at the *HSPA4* gene region. (**e**) The haplotype patterns heatmap and structure of *DNAJC1* with exons indicated by vertical bars. A missense mutation was highlighted in red.

**Figure 4 biology-11-01327-f004:**
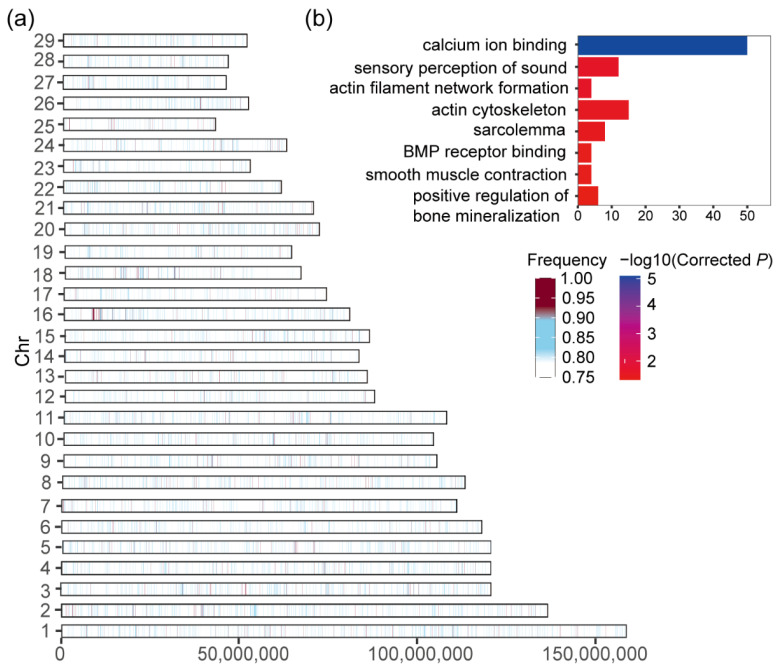
The “East Asian taurine-like” segments were identified in Dabieshan cattle. (**a**) Distribution of the “East Asian taurine-like” segments with frequencies > 0.75 for the autosomes (**b**) GO and KEGG analysis of their host gene.

## Data Availability

The datasets presented in this study can be found in online repositories. The names of the repository/repositories and accession number(s) can be found in the article/Appendix A.

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
