# Peer review of "Genetic Diversity and Selective Signature in Dabieshan Cattle Revealed by Whole-Genome Resequencing"

_biology, 2022, doi:10.3390/biology11091327_

Round 1
Reviewer 1 Report
The Authors investigated the role of any genes in Dabieshian Cattle (a particular bovine breed reed in China) and relating the genes' expression with reproductive and productive indexes.
Overall the general interest of the readers, any critical information should be included to consider the manuscript available for publication:
- English revision is required from a native English speaker
- The study presents an important limitation related to the number of animals included. This should be discussed, and the limitation put in the text. Also, the total number of Dabieshian in China should be added, as well as the number and localization of farms where these animals are reared. Do animals have bloodlines in common?
- The number of the protocol of the approval of the Ethical Committee must be included in the apposite paragraph
- The quality of the figures is very low.
Reviewer 2 Report
1. Line 84- can you provide relevant biological details of sample collection such as, samples collected at one time point or multiple, what are median age of cattle at the time of collection etc.
2. Line 133- explain briefly what is composite likelihood ratio (CLR) and their purpose here.
3. Line 154- does this mean “East Asian taurine-like” is sub-set of Dabieshan and first is not overlapping with later one, please clarify.
4. Line 210- Fig. 1- labelling is blurred, need to make them more clear for reader to read.
Reviewer 3 Report
The Authors analyzed the genetic structure, genetic diversity and genetic signature of the Dabieshan cattle.
Here are some suggestions that the Authors can consider improving the manuscript.
Line 67: please change “….scholars…..” with “….researchers …..”.
Lines 74 to 78: please reformulate the sentences. The Authors have to focus on the objective of the study and they must not anticipate "Materials and methods" (number of animals used in the study) and "Results" sections (Citation of figures).
Lines 149 to 151: please explain why the Authors performed comparison only between Dabieshan cattle and Qinchuan cattle using the FST. In my opinion, all the Chinese native cattle have to be considered.
Lines 164 to 168: the reported information are already in the “Materials and methods” section. Please reformulate the sentences and eliminate the information previously cited.
Lines 176 to 179: please move the description of the five “core” cattle populations to “Materials and methods” section (line 95).
Figures 1, 2, 3, 4: the graphical quality of the figures and the font of the text have to be improve.
Lines 344 to 348: these sentences have to be better explained.
Line 399: please change “….skeletal development. (Figure 4b).” with “….skeletal development (Figure 4b).”
Round 2
Reviewer 1 Report
well done.